# Anthocyanins Prevent AAPH-Induced Steroidogenesis Disorder in Leydig Cells by Counteracting Oxidative Stress and StAR Abnormal Expression in a Structure-Dependent Manner

**DOI:** 10.3390/antiox12020508

**Published:** 2023-02-17

**Authors:** Jun Hu, Xusheng Li, Naijun Wu, Cuijuan Zhu, Xinwei Jiang, Kailan Yuan, Yue Li, Jianxia Sun, Weibin Bai

**Affiliations:** 1Department of Food Science and Engineering, Institute of Food Safety and Nutrition, Guangdong Engineering Technology Center of Food Safety Molecular Rapid Detection, Jinan University, Guangzhou 510632, China; 2School of Chemical Engineering and Light Industry, Guangdong University of Technology, Guangzhou 510006, China

**Keywords:** anthocyanin, AAPH, oxidative stress, testosterone, R2C, steroidogenic acute regulatory protein

## Abstract

Testosterone deficiency may increase the risk of sexual dysfunction and the failure of spermatogenesis. Oxidative stress that is derived from the destruction of homeostasis, disease, and exposure to contaminants can damage the steroidogenicity process in Leydig cells, resulting in a reduction in testosterone synthesis. Anthocyanins are a group of innoxious antioxidants widely recognized in food sources, and are an ideal candidate to relieve oxidative stress-related steroidogenesis disorder. However, there is still a major gap in our knowledge of the structure–function relationship of anthocyanin on the activity mentioned above. In the present study, four anthocyanins including cyanidin-3-glucoside (Cy-3-glu), delphinidin-3-glucoside (Dp-3-glu), pelargonidin-3-glucoside (Pg-3-glu), and cyanidin-3,5-diglucoside (Cy-3,5-diglu) were applied to reverse testosterone generation after employing 2,2′-Azobis(2-amidinopropane)-dihydrochloride (AAPH) as the inducer of oxidative stress in R2C cells. The results demonstrated that all four kinds of anthocyanins can inhibit ROS generation, alleviate mitochondrial membrane potential damage, and contribute to increased testosterone. Among them, Cy-3,5-diglu with diglycoside performed best on antioxidative ability and improved cell dysfunction and upregulated the expression of the steroidogenic acute regulatory protein (StAR). The molecular docking further revealed the direct combination between anthocyanins and StAR, suggesting that anthocyanins with monosaccharide were more likely to interact with StAR than with diglycoside. Taken together, these data indicate that recipient R2C cells under oxidative stress submitted to anthocyanins exhibited improved steroidogenesis in a structure-dependent manner. Anthocyanins could be considered the ideal ingredients against oxidative stress-induced testosterone deficiency.

## 1. Introduction

Testosterone, mainly produced by Leydig cells in the testes under the stimulation of luteinizing hormone in males, is the critical prerequisite to maintain the development of male secondary sexual characteristics, spermatogenesis, and even systemic metabolism [1,2]. Testosterone deficiency can undesirably cause sexual dysfunction and semen quality reduction, finally resulting in male infertility [3]. In light of factors including environmental contaminants, improper diet, and unhealthy lifestyle, testosterone deficiency already has a prevalence of up to 20% in adolescents and young adult males in the USA, and has developed into a global issue with a prevalence between 10 and 40% [4,5]. Thus, the defined male hypogonadism attributed to testosterone deficiency severely threatens our population security. Although traditional testosterone therapy for testosterone deficiency can rapidly elevate the testosterone level, severe side effects have also been widely stated including worsening sleep apnea, stimulating noncancerous growth of the prostate, limiting sperm production, and causing testicular atrophy [6]. 

Oxidative stress is considered a chronic detrimental risk leading to dysfunction in testosterone generation [7,8]. Supraphysiologic levels of reactive free radicals (ROS) attacking the male reproductive system including superoxide anions (O2·^−^), hydrogen peroxide (H_2_O_2_), peroxyl (·ROO), and hydroxyl (·OH) are mostly from endogenous sources such as mitochondrial dysfunction and sperm membrane in immature sperm and leukocytes, and exogenous sources including smoking, alcohol intake, radiation, and environmental sources [9]. On the contrary, moderate ROS are necessary for homeostasis such as sperm capacitation [9]. ROS accumulation attributed to ROS over-production or endogenous antioxidant enzyme over-consumption is mostly correlated with damaged properties such as cell dysfunction, DNA damage, mitochondria damage, lipid peroxidation, and cell apoptosis [10]. Mounting evidence illustrates that the oxidative stress resulting from multiple incentives contributes to Leydig cell dysfunction and limited steroidogenesis. For example, oxidative stress associated with pubertal obesity caused pathological damage to Leydig cells and low testosterone [11]. The normally attendant contaminants from food sources such as cadmium [12], acrylamide [13], and chloropropanol [14] also primarily contribute to impaired steroidogenesis by oxidative stress. Briefly, the mechanism of oxidative stress-induced steroidogenesis failure is derived from damaged mitochondria and the following aberrant proteins involved in testosterone synthesis, principally including steroidogenic acute regulatory protein (StAR), cholesterol side-chain cleavage enzyme (P450scc), 17β-hydroxysteroid dehydrogenases (17β-HSD), and 3β-hydroxysteroid dehydrogenases (3β-HSD) [9,15]. Innoxious phytochemicals with antioxidation have attracted substantial attention for hormone regulation and male infertility treatment.

Anthocyanins are pigments frequently found in edible fruits, vegetables, and grains [16,17,18]. Since they present excellent anti-oxidative activity, anthocyanins, as a special dietary supplementation, have been widely considered as a proper strategy to scavenge free radicals in vivo, performing an increasingly important role in improving aging, obesity, diabetes, and vision protection [19,20]. Recently, our study found that anthocyanins are responsible for the protection of testis and spermatogenesis when confronting the threat of oxidative stress [21]. We also have reported that anthocyanins can release the oxidative stress induced by exogenous pollutants on Leydig cells in vitro, and massively enhance the process of steroidogenesis [12,14]. However, more than 700 anthocyanins have been identified in nature [22], but bioactivity in a structurally dependent manner, considering the oxidative stress injured Leydig cells, has not been well investigated. 

Herein, we include the common anthocyanins containing cyanidin-3-glucoside (Cy-3-glu), delphinidin-3-glucoside (Dp-3-glu), pelargonidin-3-glucoside (Pg-3-glu), and cyanidin-3,5-diglucoside (Cy-3,5-diglu) (Figure 1) for exploring the impact of the number of hydroxyl groups on ring B and substitutional glucoside on bioactivity against oxidative stress in R2C Leydig cells. This study aimed to investigate the structure–activity relationship of anthocyanins in reversing the oxidative stress-induced dysfunction of Leydig cells and provide a more ideal anthocyanin candidate.

## 2. Materials and Methods

### 2.1. Materials and Reagents

AAPH was purchased from J&K Scientific (Beijing, China), and different anthocyanins including Pg-3-glu, Dp-3-glu, Cy-3-glu, and Cy-3,5-diglu were purchased from Chengdu Must Bio-Technology Co., Ltd. (Chengdu, China), and DPPH was purchased from Tokyo Chemical Industry Co., Ltd. (Tokyo, Japan). The primary antibody for 3β-HSD was purchased from Abcam (Cambridge, UK). Antibodies for StAR and GAPDH were obtained from Cell Signaling Technology (Boston, MA, USA). The secondary anti-rabbit IgG antibody was from Proteintech (Wuhan, China). All other chemicals in this study were of analytical grade unless otherwise stated. 

### 2.2. Cell Culture and Treatments

R2C Leydig cells (China Center for Type Culture Collection) were maintained at 37 °C in 5% CO_2_ in DMEM/F12 medium (Life Technologies, Carlsbad, CA, USA) containing 3% fetal bovine serum (Gibco, Carlsbad, CA, USA) and 15% horse serum (Gibco, Carlsbad, CA, USA). When R2C cells were cultured to the logarithmic growth phase, they were attached to the plates for the next treatment. The frozen cells were thawed in due time to ensure the cell passage was not more than ten generations.

### 2.3. Cell Viability Determination

The cells were equably seeded in a 96-well plate at a density of 4 × 10^3^ cells per well for 24 h. Then, the medium was replaced by fresh medium containing AAPH at different concentrations of 0, 1, 2, 3, 4, and 5 mM. After 24 h, the culture medium was discarded, and 100 μL serum-free DMEM/F12 medium and 10 μL Cell Counting Kit-8 (CCK8) reagent (Beyotime, Shanghai, China) were added to each well. Afterward, the plate was placed in an incubator for 2 h at 37 °C protected from light, and the Optical Density (OD) value of each well was measured with a microplate reader (Tecan, Männedorf, Switzerland). Combined with the progesterone change and cell viability, AAPH at 160 μM further served as the damage concentration. Similarly, the adherent cells were treated with 160 μM AAPH and 50 μM different anthocyanins (Pg-3-glu, Dp-3-glu, Cy-3-glu, and Cy-3,5-diglu) for 24 h, and the cell viability was measured as previously described. 

### 2.4. Measurement of Progesterone

R2C cells were placed in a 96-well plate at a density of 4 × 10^3^ cells per well for 24 h. Subsequently, the cells were subjected to the freshly prepared medium containing different concentrations of AAPH at 0, 10, 20, 40, 80, 160, and 320 μM. After culturing for another 24 h, the medium supernatant was collected to quickly measure the progesterone content using a progesterone radioimmunoassay kit (Beijing North Institute of Biotechnology) according to the provided instruction. For further exploration, the AAPH at 160 μM and Cy-3-glu at 0, 12.5, 25, 50, 75, and 100 μM was employed to investigate the progesterone-adjusting effect from anthocyanins, and the final comparison of four types of anthocyanins (Pg-3-glu, Dp-3-glu, Cy-3-glu, and Cy-3,5-diglu) was conducted at the dosage of 50 μM, using the method mentioned above.

### 2.5. ROS Determination

The ROS levels in R2C cells were measured using a reactive oxygen species assay kit (Beyotime Biotech, Shanghai, China). Briefly, cells were placed in a 96-well plate at a density of 4 × 10^3^ per well and incubated for 24 h. After aspirating the medium and washed by fresh medium twice, a final concentration of 10 μM H2DCFDA (DCFH-DA) was added, and then incubated for 20 min. The solution was immediately removed and the well was washed three times with phenol red-free and serum-free medium, and then the fresh medium was added and the fluorescence intensity was measured with a fluorescent microplate reader (Thermo Fisher Scientific, Waltham, MA, USA) under excitation wavelength at 488 nm and emission wavelength at 525 nm, respectively. In the treated group, cells were subjected to 160 µM of AAPH and 50 µM of different anthocyanins for the standard steps with the incubation time of 5, 10, 40, 80, 160, 320, and 240 min. Each experiment included five replications.

### 2.6. Cell Mitochondrial Membrane Potential Assay

R2C cells at a density of 3 × 10^5^ cells per well were seeded in 6-well plates and cultured in a 37 °C, 5% CO_2_ incubator for 24 h. The medium was replaced with 160 μM of AAPH and 50 μM of different anthocyanins. After 24 h, the cells were digested and resuspended with 0.5 mL of medium. According to the instructions of the mitochondrial membrane potential detection kit (Beyotime Biotechnology, Shanghai, China), 500 μL of staining working solution was added to each tube, the tube was then incubated at 37 °C for 20 min, centrifuged at 4 °C, 600× *g* for 5 min, and the supernatant was discarded. Then, 1 mL of staining buffer was added to each group and centrifuged under the same conditions, and the supernatant was discarded and repeated again. Cells were resuspended in a 500 μL staining buffer, and the changes in MMP were detected by flow cytometry (Beckman Coulter, Brea, CA, USA). The red fluorescence excitation wavelength was 585 nm, emission wavelength was 590 nm; and the green fluorescence excitation wavelength was 514 nm, emission wavelength was 529 nm. Data were analyzed with Flowjo software (B&D Biosciences, Franklin Lakes, NJ, USA).

### 2.7. Protein Extraction and Western Blot

R2C cells at a density of 3 × 10^5^ cells per well in a 6-well plate were incubated for 24 h with 160 μM of AAPH and 50 μM of different anthocyanins, and then fully lysed with RIPA lysis solution (Beyotime Biotech, Shanghai, China) containing a mixture of protease inhibitors (Biosharp, Hefei, China) and phenylmethanesulfonyl fluoride (Beyotime Biotech, Shanghai, China) at the final concentration of 1 mM. Supernatants were collected after centrifugation at 13,000× *g*, 4 °C for 10 min, and protein concentrations were measured by BCA protein assay kit (Beyotime Biotechnology, Shanghai, China). Then, 2.0 g/L protein samples for separation were prepared with PBS and protein loading buffer. A total of 20 μg of protein in each sample was fractionated by 10% sodium dodecyl sulfate-polyacrylamide gel (SDS-PAGE) and electrophoretically transferred to PVDF membranes. Samples were subjected to Western blot analysis following standard protocols performed in our previous study [21], with antibodies targeted to StAR and 3β-HSD.

### 2.8. Antioxidant Properties Assessment

Different anthocyanins at 50 µM were considered for in vitro antioxidant properties assessment, and the same concentration of vitamin C was used as the positive control. DPPH free radical scavenging test, ABTS free radical scavenging test, and FRAP value test were referenced to the methods as described in our previously published study [23].

### 2.9. Molecular Docking

The predicted structure for StAR, whose UniProtKB accession numbers are P51557, was downloaded from AlphaFold Protein Structural Database and used as the receptors in the docking analysis. The 3D structures of anthocyanins were drawn using the Chem3D Pro 14.0 (Waltham, MA, USA), and their energy was minimized. The pdbqt files for the proteins and anthocyanins were created with AutoDock Tools-1.5.7 (La Jolla, CA, USA). AutoDock 4.2.6 (La Jolla, USA) was used for the molecular docking of proteins and anthocyanins. The optimal results with the maximum number of binding and the lowest binding energy were chosen, and PyMol 2.5.2 (New York, NJ, USA) was used to obtain and generate the protein–anthocyanin complex outputs.

### 2.10. Data Analysis

All tests were performed at least in triplicate. The statistical analysis of changes in color characteristics, color values of anthocyanins, anthocyanin contents, and anti-oxidative capacity was carried out using one-way ANOVA with Duncan’s multiple comparisons test as a post hoc comparison using SPSS 25.0 (Chicago, IL, USA). Data were shown as mean ± standard deviation, and statistical significance was considered when *p* < 0.05 with different letters. The data were visualized by Graph Pad Prism 8.0 (San Diego, CA, USA).

## 3. Results

### 3.1. AAPH Decreased Progesterone Level in R2C Cells in a Dose-Dependent Manner

Since AAPH produces reactive oxygen radicals under physiological conditions, it has been frequently used to cause undesired oxidative stress in an increasing number of studies. Primarily, to explore the appropriate dosage of AAPH to interfere with the secretion of progesterone without severe cell cytotoxicity, we treated R2C cells with 1, 2, 3, 4, and 5 mM AAPH for 24 h to determine the toxicity of AAPH on cell viability. It was clearly shown that AAPH less than 1 mM promised normal cell viability (Figure 2A) and cell morphology (Figure 2B), suggesting that AAPH within the concentration range of 0–1 mM could be employed in the subsequent experiment. We continued to administrate R2C cells with AAPH of 0, 10, 20, 40, 80, 160, and 320 μM for 24 h to perform whether a low dose of AAPH could be deleterious to progesterone production. As shown in Figure 2C, a dose-dependent manner was represented, in which the progesterone level in the medium was negatively related to the increased AAPH concentration. Concretely compared with the control group, the progesterone massively decreased by 44% at 160 μM AAPH. Therefore, 160 μM of AAPH was considered in the further study.

### 3.2. Anthocyanins Improved AAPH-Induced Decrease in Progesterone Synthesis

Cyanidin-3-O-glucoside (Cy-3-glu) is the major dietary anthocyanin and frequently exists in the plant world. Thus, Cy-3-glu was foremost taken as the ingredient candidate to eradicate the progesterone dysregulation stimulated by AAPH. After evaluating the progesterone level in the supernatant submitted to anthocyanins in the AAPH model, Cy-3-glu partially rendered the damaged cells for regression of progesterone-producing capacity when the dosage of intervention was from 25 to 75 μM (Figure 3A). According to the above results, the concentration of anthocyanins was selected as 50 μM for the structure–function relationship investigation. Then, Pg-3-glu, Cy-3,5-diglu, Dp-3-glu, and Cy-3-glu with a concentration of 50 μM firstly confirmed the non-cytotoxicity in the AAPH model (Figure 3B). Subsequently, the recipient cells submitted to AAPH and various anthocyanins displayed discrepancies in the observed effects of progesterone enhancement. Based on the results expressed in Figure 3C, Cy-3,5-diglu was the most desirable anthocyanin to intensify progesterone secretion compared to Pg-3-glu, Dp-3-glu, and Cy-3-glu.

### 3.3. Anthocyanin Repaired Oxidative Stress Damage in R2C Cells

To elaborate on the mechanism of anthocyanin function and structure–response relationship, we focused on the predictable oxidative stress damage induced by AAPH based on the mechanism involved in reactive oxygen species formation. As shown in Figure 4A, after AAPH intervened in the cells, the intracellular ROS level continued to increase over time, and correspondingly, diverse anthocyanins showed different degrees of eradication of ROS retention after repairment, which was lower than that of the control group and showed the attenuated ability as following Cy-3-glu, Cy-3,5-diglu, Dp-3-glu, and Pg-3-glu. We next conducted the in vitro anti-oxidative capacity examination including DPPH free radical scavenging, ABTS free radical scavenging, and FRAP value test. Consistent with the ROS elimination in R2C cells, Cy-3-glu and Cy-3,5-diglu normally exhibited better anti-oxidative ability compared to other anthocyanins and even better than the same dose of Vitamin C (Figure 4B–D). These data indicated that anthocyanins indeed engaged in the re-establishment of the oxidative microenvironment associated with the expenditure of ROS. 

### 3.4. Anthocyanins Reduced Mitochondrial Damage 

Mitochondrial dysfunction is highly correlated to hormonogenesis disorder since mitochondria are the indispensable factory for progesterone exportation, and cells exposed to oxidative stress damage are normally subjected to a decline in mitochondrial membrane potential. We characterized the role of different anthocyanins on the mitochondrial membrane potential (MMP) adjustment. After AAPH treatment, the proportion of cells with decreased MMP was significantly increased compared to the control group, indicating mitochondrial damage (Figure 5). Anthocyanins with different structures all have shown the relieving of oxidative stress, and the proportion of damaged cells with increased MMP was visibly reduced. Similar to the abovementioned data, Cy-3,5-diglu was also more efficient in protecting the AAPH-damaged cells in the aspect of MMP. Collectively, anthocyanins serve as a safeguard on the mitochondrial function of R2C cells. 

### 3.5. Anthocyanins Restored the Expression of StAR for Normal Progesterone Synthesis in Damaged R2C Cells

StAR and 3β-HSD are necessarily required for progesterone synthesis in vivo and in vitro. Particularly, StAR is responsible for the transport of the starting material cholesterol and the facilitated conversion of cholesterol to pregnenolone by the P450_scc_ enzyme system, while 3β-HSD is posited as the pivotal protein at the final step of progesterone synthesis. Herein, we conducted a protein immunoblotting to quantify the protein expression of StAR and 3β-HSD to determine whether the pathway involved in progesterone production is influenced by AAPH-induced oxidative damage and the corresponding effect of anthocyanins (Figure 6A). The results were also highly consistent with previous experiments. After 24 h of AAPH treatment, the expression of StAR was significantly suppressed compared with the control group, demonstrating that AAPH-caused mitochondrial damage can perturb the initiation of progesterone synthesis. Nevertheless, after the intervention of anthocyanins with different structures, this negative effect can be improved. It is worth noting that although anthocyanins can initially relieve oxidative stress, the StAR improvement assisted by anthocyanins in different structures is not totally concomitant with the changing trend, considering the diverse antioxidative ability as described previously (Figure 6B). Cy-3-glu expressed desirable in vitro anti-oxidative capacity as well as the best effect on ROS quenching during cell culture; however, only fine-tuning was observed on the StAR expression (Figure 6C). Thus, we hypothesized a supererogatory regulatory effect was directly contributed by anthocyanins in structural dependence. After conducting molecular docking using four types of anthocyanins and StAR protein (Figure 6D), the binding energy revealed that the Cy-3,5-diglu–StAR complex possessed the lowest energy, which the binding energy forming an anthocyanin–StAR complex of Cy-3,5-diglu, Cy-3-glu, Pg-3-glu, and Dp-3-glu is −3.18, −4.38, −4.49, and −4.36 Kcal/mol, respectively (Table 1). The docking simulations demonstrated that Cy-3,5-diglu is inclined to selectively bind to the Asn103 and ASP126 residues of the StAR active site pocket through hydrogen-bond interactions. Meanwhile, the Pg-3-glu targets to Ile244, Asn256, and Leu246; Cy-3-glu targets to Asp245, Trp249, Lys252; and Dp-3-glu targets to Gly246, Asn256, Lys117, and Leu250. Regarding the expression of 3β-HSD, no significant change was observed during the treatment. These data indicate that AAPH-induced mitochondrial injury restricted progesterone synthesis through repressing StAR expression, but anthocyanins participated in the protection of StAR in structural dependence.

## 4. Discussion

Along with the development of the high quality of human life, various environmental pollutants, drug abuse, and overnutrition have gradually backfired on human health [24,25]. Especially for the male reproductive system, these exogenous harmful substances not only cause oxidative stress damage but can also lead to the decline of reproductive function in the long term. Oxidative stress damage to the reproductive system simultaneously leads to a decrease in testicular mesenchymal cell enzyme activity and hormonogenesis disorder. Anthocyanins are a group of water-soluble natural pigments commonly found in dark fruits and vegetables in the daily diet and involved in excellent bioactivities [19]. In our previous study, anthocyanin Cy-3-glu showed improvement in cadmium-induced sex hormone dysfunction in vitro and in vivo [12,26]. However, differences remain in the physicochemical properties of anthocyanins with different structures; in addition, the mechanism of anthocyanin production in the protection of progesterone in the male reproductive system is not clear. Therefore, we chose AAPH as a modeling substance for oxidative stress injury in R2C cells and then intervened with four anthocyanins to investigate the specific mechanism of anthocyanin protection and the effect of structure on this efficacy.

The Leydig cell that is anchored to the testicular interstitial is responsible for steroidogenesis, maintaining critical phases of development and homeostasis of male physiological functions. Testosterone synthesis in Leydig cells is a progressive process that converts cholesterol into final testosterone, which requires various critical enzymes such as StAR, P450scc, 3β-HSD, CYP17A1, and 17β-HSD. A growing body of evidence has documented that exogenous and endogenous oxidative stress are extremely connected with Leydig cell dysfunction accompanied by decreased progesterone and testosterone secretion. Frequently, endocrine-disrupting chemicals are always capable of altering the redox environment to increase oxidative stress, following suppressed steroidogenesis. Our previous research also proved that Leydig cells exposed to cadmium and chloropropanol will suffer a supraphysiologic level of ROS and, finally, cause mitochondrial damage [12,14]. Other exogenous free radicals produced by local lesions from other organs under disease conditions such as diabetes [27], obesity [11], and non-alcoholic fatty liver disease [28] will also contribute to the dysfunction of Leydig cells. More importantly, as an intrinsic factor, endogenous ROS appeared during steroidogenesis, including in the mitochondrial electron transport chain, mitochondrial and microsomal cytochrome P450 enzyme reactions, and luteinizing hormone-dependent pathway [29,30]. Moreover, age-related reduction in the antioxidant defense system of Leydig cells also contributes to the decline of androgen. Therefore, preventing the Leydig cells from chronic oxidative stress by providing antioxidants such as anthocyanins is of great potential to ameliorate steroidogenesis disorder. AAPH is a classical compound that decomposes spontaneously to generate multiple free radicals during cell culture and is approximated to real status in vivo. The R2C cells were submitted to a low dosage of AAPH without severe cell damage and reduction in cell viability, considering that chronic oxidative stress in vivo generally does not lead to extensive cell loss. However, further results showed that even AAPH at a fairly low concentration associated with ROS generation has already given rise to the deterred progesterone production.

Anthocyanins are normally regarded as excellent antioxidants in nutritional supplementation and disease prevention [19]. There are six common natural anthocyanin aglycons including pelargonidin, cyanidin, delphinidin, peonidin, petunidin, and malvidin based on the variant structure of the parent ring. The distributed phenolic hydroxyl groups pertaining to the anti-oxidative ability cause heterogeneity in bioactivity and stability in different anthocyanins. In the present study, we considered four types of anthocyanins involving Pg-3-glu, Cy-3-glu, Dp-3-glu, and Cy-3,5-diglu, which are different in the number of phenolic hydroxyl groups at Ring B and the number of glycosides. In vitro study expresses the best anti-oxidative ability originating from Cy-3-glu and Cy-3,5-diglu. This capacity can be attributed to the catechol structure on Ring B in Cy-3-glu and Cy-3,5-diglu, and it has been fully illustrated that catechol can provide better stability and antioxidant activity by forming benzoquinone. Moreover, o-phenylene triol at Dp-3-glu is reported as not stable enough to persistent existence in a physiological microenvironment [31]. As consequence, ROS accumulation during cell culture was largely removed in the presence of Cy-3-glu and Cy-3,5-diglu. The excessive reduction in the anthocyanin-treated group compared to the control group may be from the extra elimination of endogenous free radicals during the test. Regardless of being endogenous or exogenous, ROS has the propensity to attack the mitochondria associated with DNA, protein, and lipid damage following progressive cell dysfunction [32]. In the present study, an increase in mitochondrial membrane potential suggests that the expected mitochondrial damage is a consequence of ROS accumulation. However, consistent with the released oxidative stress, R2C cells subjected to Cy-3,5-diglu conferred a more remarkable reduction in the number of damaged cells. Furthermore, we observed that the progesterone reduction triggered by oxidative stress was dissipated after anthocyanins treatment. Uniformly, Cy-3,5-diglu also performed the most effective reversion compared to the others three anthocyanins. Consequently, these data together revealed the positive relationship between the oxidation resistance of anthocyanins and their protective effects on oxidative stress-induced R2C cell dysfunction. 

The unique compartmentalization of mitochondrial membranes is developed to regulate steroid synthesis in steroidogenic cells [33]. It could be speculated that the downstream synthesis of progesterone in R2C cells was undoubtedly confined under oxidative stress, considering the already damaged mitochondrial associated with destabilized MMP. As the rate-limiting step, the cholesterol should be transferred into the mitochondria by StAR as a prerequisite for progesterone formation [26]. Therefore, we validated that StAR protein expression was significantly decreased in the presence of AAPH. Cy-3,5-diglu exhibited profound effects on the upregulation of the StAR level. Thus, the function of Cy-3,5-diglu may partially benefit from the proposed anti-oxidative performance and render the cells free from incidental damage. However, aside from Cy-3,5-diglu, other anthocyanins including Pg-3-glu, Dp-3-glu, and Cy-3-glu all have no contribution to StAR improvement. This ineffectiveness of StAR regulation is inconsistent with the previous result that three monosaccharide anthocyanins all swept the induced ROS and increased the final progesterone level. Regarding the limitation of ROS detection and the distinctly observed difference between these three anthocyanins-treated groups and the control group in terms of the MMP damage ratio, we assumed that these monosaccharide anthocyanins can bind directly to the StAR. Subsequently, the molecular docking indicated that monosaccharide anthocyanins dramatically prevail over Cy-3,5-diglu in binding energy for forming an anthocyanin complex. However, the biofunction after combination at different amino acid sites is unclear, and there is also a major gap in our knowledge of the activation of the small molecule–StAR complex. Additionally, as the fundamental determinant of anthocyanin stability, increasing glycosylation imparts greater stability to anthocyanins [34]. Because the anthocyanin degradation that was undergone during cell culturing extremely influences bioavailability, the changed anthocyanin expenditure may finally contribute to the observed differential cell response at the protein level. 

Together, anthocyanins protected the oxidative stress-induced steroidogenesis dysfunction of R2C cells in a structure-dependent manner. The structural difference in Ring B and glycoside played a critical role in the protective effects. Among these four anthocyanins, Cy-3,5-diglu possessed a catechol structure and had the optimum anti-oxidative ability, minimal MMP damage, and strongest regulation in strengthening StAR expression, eventually resulting in the largest promotion in progesterone secretion. In the contrast, Cy-3-glu, which is distinguished from Cy-3,5-diglu in glycoside, although expressed the superfluous ROS elimination, had no contribution to StAR regulation. Thus, Cy-3-glu is not as effective as Cy-3,5-diglu in improving progesterone levels. Dp-3-glu and Pg-3-glu are not appropriate to reverse the oxidative stress-induced Leydig cells dysfunction due to the lack of anti-oxidative ability and StAR adjustment, considering the unstable o-phenylene triol and the single phenolic hydroxyl group at Ring B, respectively. As a limitation of this study, further study should draw substantial attention to the interaction between anthocyanins and active proteins to reinforce the more efficient anthocyanins targeted to steroidogenesis. Moreover, the screening of different anthocyanins against ROS-induced steroidogenesis dysfunction was supposed to not only focus on the antioxidation but also on the direct StAR binding and regulation.

## 5. Conclusions

In summary, AAPH caused a series of damage to R2C cells, which ultimately affected progesterone synthesis by inhibiting StAR, a key protein for progesterone synthesis. This damage could be ameliorated by different structural anthocyanin. The protective effect of anthocyanins against the oxidative damage of R2C cells caused by AAPH was positively correlated with the antioxidant capacity of the anthocyanins, when considering Cy-3-glu, Cy-3,5-diglu, Pg-3-glu, and Dp-3-glu as the ideal ingredients. All four anthocyanins can significantly improve progesterone secretion compared to damaged cells, and the effects highly also depend on the discrepant structure when considering StAR regulation. Moreover, the catechol structure in Cy-3,5-diglu has stronger antioxidant properties and performs ameliorative mitochondrial damage and StAR expression. These data suggest that anthocyanins are of great potential to protect the oxidative stress-induced steroidogenesis disorder and can be considered as a promising applicable strategy to alleviate the detrimental effects.

## Figures and Tables

**Figure 1 antioxidants-12-00508-f001:**
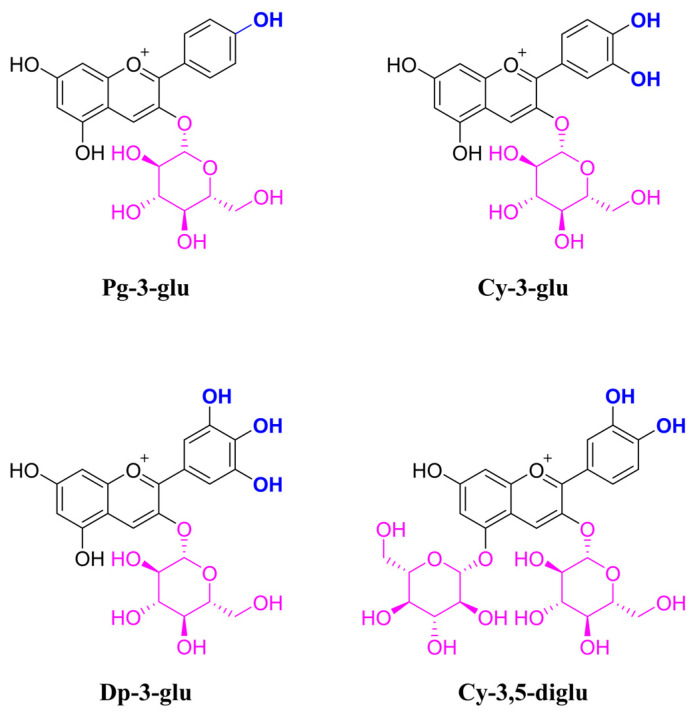
Molecular structures of four anthocyanins, Pg-3-glu, Cy-3-glu, Dp-3-glu, and Cy-3,5-diglu.

**Figure 2 antioxidants-12-00508-f002:**
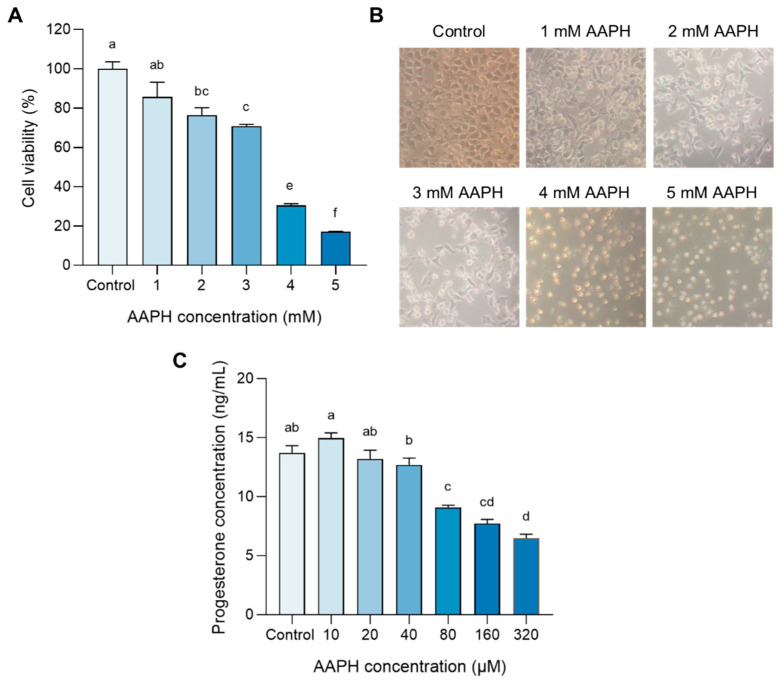
Effect of AAPH on R2C cell viability and progesterone synthesis: (**A**) Cell viability of R2C cells under different concentrations of AAPH, n = 4. (**B**) Representative cell morphology of R2C cells treated with 1, 2, 3, 4, and 5 mM AAPH. (**C**) Progesterone showed a dose-dependent decrease in AAPH concentration within 320 μM, n = 6. All data were analyzed for differences by one-way ANOVA. Different letters indicated that there is a significant difference (*p* < 0.05).

**Figure 3 antioxidants-12-00508-f003:**
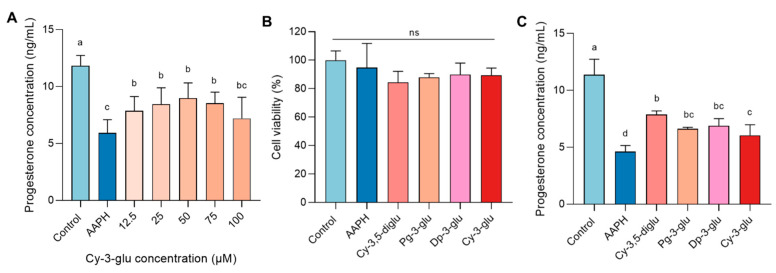
Anthocyanins ameliorate progesterone secretion in AAPH-damaged R2C cells: (**A**) Recovery of progesterone synthesis by different concentrations of Cy-3-glu, n = 6. (**B**) Effect of four anthocyanins on the viability of R2C cells, n = 5. (**C**) The change in AAPH-induced reduction in progesterone synthesis by 50 μM of four anthocyanins, n = 4. All data were analyzed for differences by one-way ANOVA. Different letters indicated that there is a significant difference (*p* < 0.05).

**Figure 4 antioxidants-12-00508-f004:**
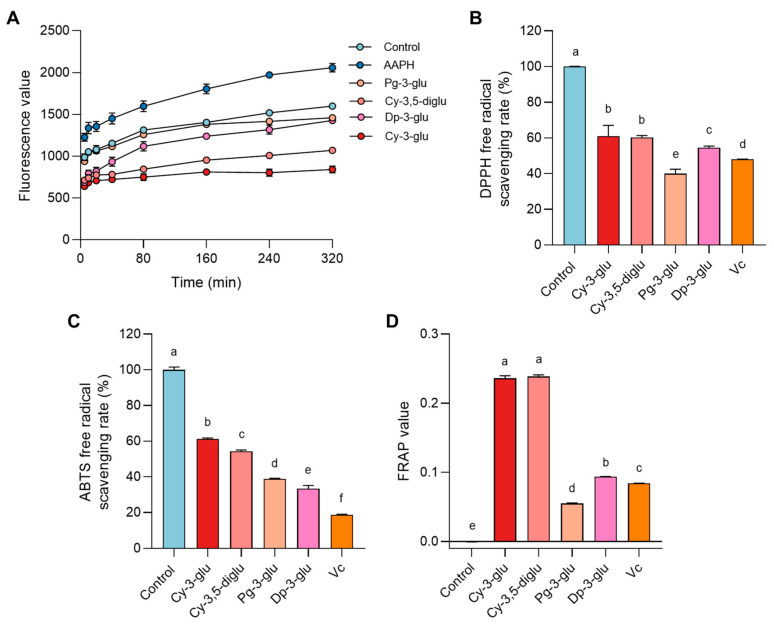
The antioxidant capacity of anthocyanin can improve cellular oxidative stress damage: (**A**) The change in ROS level after treatment with 160 µM of AAPH and 50 µM of anthocyanins, n = 5. Measurement of in vitro antioxidant capacity of 50 µM of different anthocyanins. (**B**) DPPH, n = 3. (**C**) ABTS, n = 3. (**D**) FRAP, n = 3. All data were analyzed for differences by one-way ANOVA. Different letters indicated that there is a significant difference (*p* < 0.05).

**Figure 5 antioxidants-12-00508-f005:**
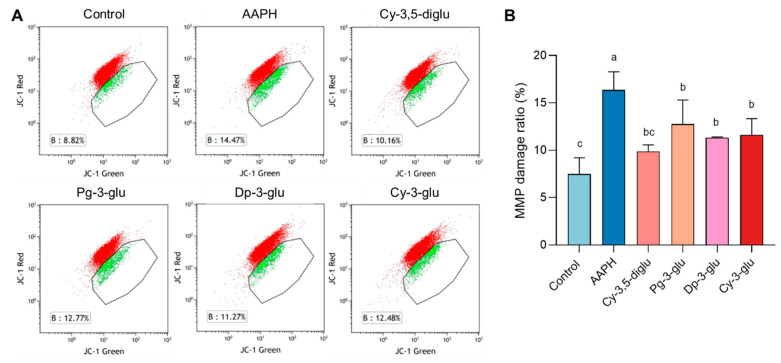
Anthocyanins reduce mitochondrial membrane potential damage: (**A**) Representative flow cytometry diagram of changes in MMP after treatment with four anthocyanins. (**B**) The ratio analysis of MMP damage cells, n = 3. All data were analyzed for differences by one-way ANOVA. Different letters indicated that there is a significant difference (*p* < 0.05).

**Figure 6 antioxidants-12-00508-f006:**
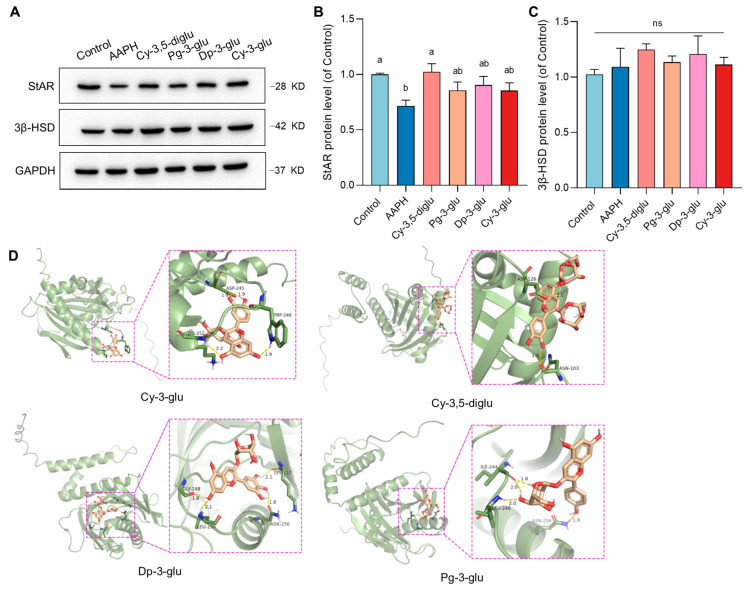
Anthocyanins target the protein involved in progesterone synthesis: (**A**) The representative photographs of Western blot band of StAR and 3β-HSD. (**B**) The grayscale analysis of StAR, n = 3. (**C**) The grayscale analysis of 3β-HSD, n = 3. (**D**) Molecular docking of four anthocyanins and StAR. Different letters indicated that there is a significant difference (*p* < 0.05).

**Table 1 antioxidants-12-00508-t001:** Binding energy and sites between four anthocyanins and StAR based on molecular docking.

Complex	Binding Energy (Kcal/mol)	Intermol Energy (Kcal/mol)	Torsional Energy (Kcal/mol)	Hydrogen Bonds between Ligands and the StAR		
				Distance (Å)	Donor Atom	Acceptor Atom
Pelargonidin-3-glucoside-StAR	−4.49	−7.77	3.28	1.8	PG3G:H38	A:ILE244:O
				1.9	A:ASN256:HD22	PG3G:O21
				2.0	A:LEU246:HN	PG3G:O34
				2.0	PG3G:H35	A:ILE244:O
Cyanidin-3-glucoside-StAR	−4.38	−7.67	3.28	1.9	C3G:H38	A:ASP245:OD1
				1.9	C3G:H40	A:ASP245:OD1
				1.9	A:TRP249:HE1	C3G:O35
				2.0	A:LYS252:HN	C3G:O33
Delphinidin-3-O-glucoside-StAR	−4.36	−8.24	3.88	1.8	DP3G:H12	A:GLY248:O
				1.8	DP3G:H26	A:ASN256:OD1
				2.1	DP3G:H22	A:LYS117:O
				2.1	A:LEU250:HN	DP3G:O11
Cyanidin-3,5-diglucoside-StAR	−3.18	−8.55	5.37	1.9	C35OG:H21	A:ASN103:O
				2.0	C35OG:H38	A:ASP126:OD2

## Data Availability

All of the data are contained within the article.

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
