# Peer review of "Anthocyanins Prevent AAPH-Induced Steroidogenesis Disorder in Leydig Cells by Counteracting Oxidative Stress and StAR Abnormal Expression in a Structure-Dependent Manner"

_antioxidants, 2023, doi:10.3390/antiox12020508_

Round 1

Reviewer 1 Report

REVIEW: Anthocyanins prevent AAPH-induced steroidogenesis disorder in Leydig cells by counteracting oxidative stress and StAR abnormal expression in structural dependent

The authors have explored the impact of anthocyanins (AC) on loss of steroidogenesis due to oxidative stress in R2C Leydig cells.  The study is fairly straightforward but despite that, there are a number of queries and comments that arise.

11.     Query the use of ‘structural dependent’ as a noun in the title

2.     Abstract: there are spelling and grammatical errors here, and the subjects of some sentences are not correctly positioned, leading to lack of clarity/confusion.

3.     The R2C cell line has been used as a Leydig cell model, leading to the possibility of cell-line specific effects.  Have the outcomes been confirmed in other cell lines, or in primary cells?

4.     The authors quote ‘prevalence’ of loss of testosterone output in the Introduction: is this global or country-specific?

5.     Despite the focus of the introduction, progesterone, rather than testosterone, output has been measured – there is the possibility that the authors have missed ‘downstream’ impact of the drugs tested.

6.     ‘And so on’ is used repeatedly – this phrase conveys nothing and any comments of this type should be removed, as should any hyperbole (e.g. masterly, momentous)

7.     Peroxyl is incorrectly identified as proxyl in the introduction

8.     What concentrations of the anthocyanins mentioned can be achieved in the bloodstream when consuming food stuffs or supplements – are they sufficient to bring about these outcomes?  The authors have typically used 50 microM?

9.     Line 72-73, 91: Sp. Errors; line 75 implies other than what the authors intend

10.  Query the statement ‘most typical’ referring to the AC: according to whom?  Without context or a supporting reference, this is an unsubstantiated statement.

11.  How physiologically relevant is AAPH as the source of oxidative stress in this cell model?

12.  Figure 1: the actual structures (not abbreviated forms) should be presented, particularly as the authors attempt to relate structure to binding function later on.

13.  All abbreviations need to be defined at first usage, throughout the manuscript (e.g. CCK8, DCFA-DH)

14.  Methods: passage numbers and limits should be defined for the R2C cells.

15.  What is the linear range of the RIA kit used for progesterone quantitation?

16.  The assay used to measure mitochondrial membrane potential: more details are needed; the same is true of the flow cytometry work.

17.  The concentrations of protease inhibitors (which?) and PMSF are needed : the reader could not repeat the work without such details; g is preferred to rpm; the % of the SDS PAGE gel should be defined; the molecular weight markers should also be indicated and shown.

18.  The authors need to confirm that the n numbers quoted in figure legends refer to independent experiments (not replicates, or combined replicates across multiple experiments).

19.  Line 224: a reference is needed to support the authors claim

20.  Line 230: over-interpretation of the data. The authors cannot infer what might happen in an individual consuming these molecules, either in food or supplements.  The most that can be claimed is a ‘bell-shaped’ response.

21.  The controls are missing from Fig 3C: although we can see there is no change in viability (3B), we cannot assume these molecules do not impact on progesterone production in the absence of AAPH.  This is important as AAPH does not affect viability at the concentration indicated but clearly does alter hormone output; the same may be true for the AC.

22.  The concentrations of AC in Figure 4 need to be indicated in the legend; the data has also been normalised – an indication of the variability of the control group is required.

23.  The Western blot for StAR: no phospho-StAR has been detected, indicating that progesterone output may not be optimal under the conditions tested in this model.

Reviewer 2 Report

This study investigated the preventive effect of anthocyanins against oxidative stress in Leydig cells via StAR expression. The experiments were well-performed, and the results were clear to conclude the authors’ claim. However, the model that the treatment of anthocyanins such as C3G prevents oxidative stress and MMP damage, leading to upregulating the expression of StAR in Leydig cells has already been claimed in the authors’ previous work (Li et al., 2019). Therefore, I consider that this work's novelty seems to be identifying which anthocyanin is more effective against oxidative stress and the structural-basis explanation by the informative simulation. Then, the authors should modify the discussion and conclusion to emphasize the novelty of this study more. Additionally, I think the authors should discuss the relationship between the effect of steroidogenesis in Leydig cells against oxidative stress and the structural information of each anthocyanin.
